# Characterizing Variances of Adulterated Extra Virgin Olive Oils by UV-Vis Spectroscopy Combined with Analysis of Variance-Projected Difference Resolution (ANOVA-PDR) and Multivariate Classification

Boyan Gao 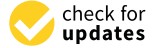, Jingyao Zhang and Weiying Lu *

Institute of Food and Nutraceutical Science, Department of Food Science and Technology,
School of Agriculture and Biology, Shanghai Jiao Tong University, Shanghai 200240, China
* Correspondence: weiying.lu@sjtu.edu.cn

**Abstract:** The analysis of variance-projected difference resolution (ANOVA-PDR) was proposed and compared with multivariate classification for its potential in detecting possible food adulteration in extra virgin olive oils (EVOOs) by UV-Vis spectra. Three factors including origin, adulteration level, and adulteration type were systematically examined by the ANOVA-derived methods. The ANOVA-PDR quantitatively presented the separation of the internal classes according to the three main factors. Specifically, the average ANOVA-derived PDRs of the EVOO origination and adulteration level, respectively, is 4.01 and 1.78, while the conventional PDRs of the three factors are all less than 1.5. Furthermore, the partial least-squares-discriminant analysis (PLS-DA) and the PLS regression (PLSR) modeling with the selected sub-datasets from different origins were used to verify the results. The resulting models suggested that the three main factors and their interactions were all important sources of spectral variations.

**Keywords:** ANOVA-PDR; extra virgin olive oil adulteration; UV-Vis spectroscopy; partial least-squares

## 1. Introduction

Olive oil is a widely used food ingredient around the world. According to the International Olive Council, the global table olive production has more than tripled in the past three decades, reaching over three million tons in the 2020–2021 crop year [1,2], with a 162% increase in consumption [1]. Despite its widespread acceptance, extra virgin olive oil (EVOO) produced in Europe, particularly in Mediterranean countries such as Spain and Italy, is considered to be of the highest quality and nutritional value. However, the table olive production of the European Union is limited, accounting for less than one-third of the world's table olive production in 2020–2021 [2]. Furthermore, European EVOOs have a higher market value, making them vulnerable to adulteration with cheaper vegetable oils such as sunflower, rapeseed, corn, or soybean oils. As a result, reliable quality assurance techniques are needed to protect consumers' interests.

According to previous studies, the UV-Vis spectroscopy combined with chemometrics is one of the important techniques for the adulteration detection, authentication of the geographic location or the grade of a specific olive oil product. For instance, Torrecilla et al. quantified the level of adulteration in Spanish EVOO from their UV-Vis spectra [3,4]. The level of adulteration was quantified using linear and nonlinear modeling based on 17 chaotic parameters calculated by UV-vis scans. Linear models with more independent variables showed better statistical results. A radial basis network model with one input node and one output neuron was used for nonlinear modeling. Jiang et al. established an effective detection model of Italian EVOO-vegetable oil combined with principal component analysis (PCA) and partial least-squares regression (PLSR) using UV-Vis [5]. Furthermore, subsequent studies also reported that UV-Vis spectroscopy was applied to establish

EVOO adulteration models [6,7]. The UV-Vis spectroscopy has been used to determine the geographic origin of EVOO as well [8,9]. In these studies, UV-Vis spectroscopy and high-performance liquid chromatography with a diode array detector were used to quantify main pigments in several EVOOs and compared the advantages and disadvantages of both techniques. The methods were applied to a selection of monovarietal EVOOs produced in different geographical areas in Mediterranean countries. The differences among EVOOs produced in different geographic areas were analyzed using principal component analysis (PCA) and independent component analysis to evaluate the correlation between pigments' content such as chlorophylls and carotenoids in olive oils and experimental factors such as ripeness stage, geographic origin, and cultivars. For brevity, "experimental factor" is addressed as factors for all subsequent descriptions. Our previous research also demonstrated that the microtiter plate reader can be utilized as a high-throughput UV-Vis spectrometer to establish an effective differentiation model for different EVOO manufacturers [10]. The advantages of UV-Vis are two-fold: both the cost is significantly lower and the sample treatment is usually simpler compared with other methods such as chromatography, infrared, and Raman spectroscopy that require either relatively expensive instruments or complex experimental procedures. However, the UV-Vis spectroscopy also has the tradeoff of relatively low selectivity and sensitivity. Therefore, to establish an effective and robust EVOO adulteration detection model, an in-depth understanding of the characteristics of the spectroscopic fingerprints under different factors is necessary.

Multi-factors can significantly impact the chemical analysis procedure, such as accuracy, sensitivity, and reproducibility. In the case of EVOO adulteration detection, factors such as origin, adulteration level, and type of the adulterant can affect the robustness of the model. Analyzing the relationships between these factors can guide the establishment of subsequent detection models and evaluate their significance. Techniques to analyze the influence of multiple factors are highly desirable for accurate analysis of EVOOs from different manufacturers.

The multivariate extensions of the analysis of variance (ANOVA), ANOVA-principal component analysis (PCA) was proposed by Harrington et al. to separate the variation of the experimental hypothesis from other sources of variation [11]. The ANOVA-PCA effectively treats the factor impacts and interactions between factors. It has been applied in determining the sources of variances in milk powder [12,13], as well as in agricultural products such as lettuce, broccoli, and dry bean, evaluating the impact of cultivar and growth conditions [12,14,15]. Additionally, the pooled-ANOVA can test the difference between two or more vectors by means of comparing the pooled variance of the variables [14,16]. The pooled-ANOVA provides a conservative test for the differences between the level averages of each factor, extending the ability of ANOVA to the multivariate domain [13].

Despite the previous successes of the various applications of ANOVA-PCA, there is no simple metric of comparing the effect of class separation under the multivariate context. The projected difference resolution (PDR) is a straightforward tool for the resolutions between groups of multivariate data objects [17]. The PDR is a single figure similar to the chromatographic resolution, so it is easy to interpret [18]. This method has been successfully applied in the authentication of cannabis [19,20], identification of rice varieties [21], etc. Analogous to ANOVA-PCA, the PDR can also be incorporated into ANOVA using the factor matrix decomposed by ANOVA. The derived methods, referred to as ANOVA-PDR, may provide useful supplemental information besides ANOVA-PCA and pooled-ANOVA.

The aim of this study was to propose a novel method, analysis of variance-projected difference resolution (ANOVA-PDR), for detecting EVOO adulteration while considering multiple influencing factors, including origin, adulteration level, and adulteration type. The UV-Vis spectra of adulterated EVOOs were comprehensively analyzed using ANOVA-PDR techniques, and the results were validated using PLS-DA and PLSR to build both quantitative and qualitative adulteration models. ANOVA-PDR can evaluate modeling performance in relation to the multiple sampling factors of the EVOO adulteration detection model.

## 2. Materials and Methods

### 2.1. Sample Pretreament

Ten commercial EVOO samples produced from five countries including Spain (S1–S4), Italy (I1–I3), Greece (G1), Portugal (P1), and Australia (A1) were purchased from local grocery stores in China. Each sample was 300–500 mL, stored in their respective original glass containers at ambient temperature, and kept sealed until analysis. Because Spain and Italy are the main producers of olive oil, the corresponding sample sets consist of four and three different manufactures, respectively. The other three EVOO origins were from the other countries to compare the possible differences between geographic location. Three commercial vegetable oils including corn, soybean, and sunflower oil were selected as possible adulterants and were purchased from local groceries.

To simulate the adulteration of EVOOs, a series of binary blended oils were prepared by adding either corn oil, soybean oil or sunflower oil into EVOOs at percentages ranging from 10% to 50% at a 10% interval (*v/v*). The samples were then vortexed for 1 min until forming a homogenous suspension. The pure and the adulterated EVOOs were directly transferred to a microtiter plate without further pretreatment. The sample volume was 200 μL for each sample. All samples were prepared in triplicates.

### 2.2. Microtiter Plate Reader Assay

All samples were placed in a Nunc MicroWell transparent 96-well plates (Thermo Fisher Scientific, Waltham, MA, USA) and analyzed at room temperature using an Infinite M1000 PRO microtiter plate reader (Tecan Group Ltd., Männedorf, Switzerland). In our previous study, it has been demonstrated that a microtiter plate reader can be a high-throughput alternative to achieve comparable performance of the benchtop counterparts [10]. The microtiter plate reader was equipped with a Quad4 monochromator and a xenon lamp. The wavelength was set to 366–1000 nm with a 2 nm resolution, and the number of flashes was 25. Each sample was prepared and tested in triplicates, resulting totally nine parallel datasets obtained for each oil sample. The final spectral dataset comprised a $1440 \times 350$ matrix, where rows and columns represent samples and variables, respectively.

### 2.3. Theory and Implementation of ANOVA-PDR

The calculation procedure of the ANOVA in a multivariate scenario is demonstrated in Figure 1. Briefly, before the ANOVA, the obtained spectral data matrix was mean-centered for each measured variable to acquire the grand means matrix. Afterwards, the original matrix was subtracted by the grand means matrix to obtain the grand residuals matrix [12]. The grand residuals matrix was then used to construct multiple sub-matrixes, i.e., the means and residuals matrix of each factor. Specifically, the factors of this EVOO adulteration study included origin, adulteration level, and adulteration type. The sub-matrices allowed calculation of the percentage of total variance for each factor, the significance level of the variance, and the variance associated with factor interactions.

The PDR is a straightforward multivariate metric for rapidly quantifying the degree of separation from multivariate data objects for a pair of classes [17]. The PDR performs multivariate resolution by generating a set of projections onto the difference vectors of two class averages between pairs of means divided by 2 times the summed standard deviations, given by

$$R_s(a,b) = \frac{\left| \overline{P_a} - \overline{P_b} \right|}{2(S_a + S_b)} \tag{1}$$

from which $R_s(a,b)$ is the PDR of class $a$ and $b$, $P_a$ and $P_b$ are obtained by projecting the objects in the corresponding category onto the difference vector between the mean of two classes, given by

$$P_i = (\overline{X_a} - \overline{X_b})X_i^T \tag{2}$$

for which $X$ is the two-way matrix for each target class of the UV-Vis dataset; thus, $X_i^T$ is the transposed two-way matrix for each target class, where $i$ stands for class $a$ or $b$. The

projection of an object $i$ on the difference vector is calculated by $\overline{X_a} - \overline{X_b}$. From these projections, the averages $\overline{P_a}$ and $\overline{P_b}$, and their corresponding standard deviations $S_a$ and $S_b$ are calculated, and, finally, the resolution between class a and class b is obtained by Equation (1). A PDR greater than 1.5 indicates that the two classes are well-resolved. The larger the PDR, the greater the resolution between the two classes. The PDRs of a dataset that contains three or more classes can be sorted in the order of an upper or lower triangular matrix with a size equal to the number of classes, where the PDR of classes $a$ and $b$ was given in the columns $a$ and rows $b$, respectively. The geometric mean of all PDRs was used as the average PDR of all classes in the dataset.

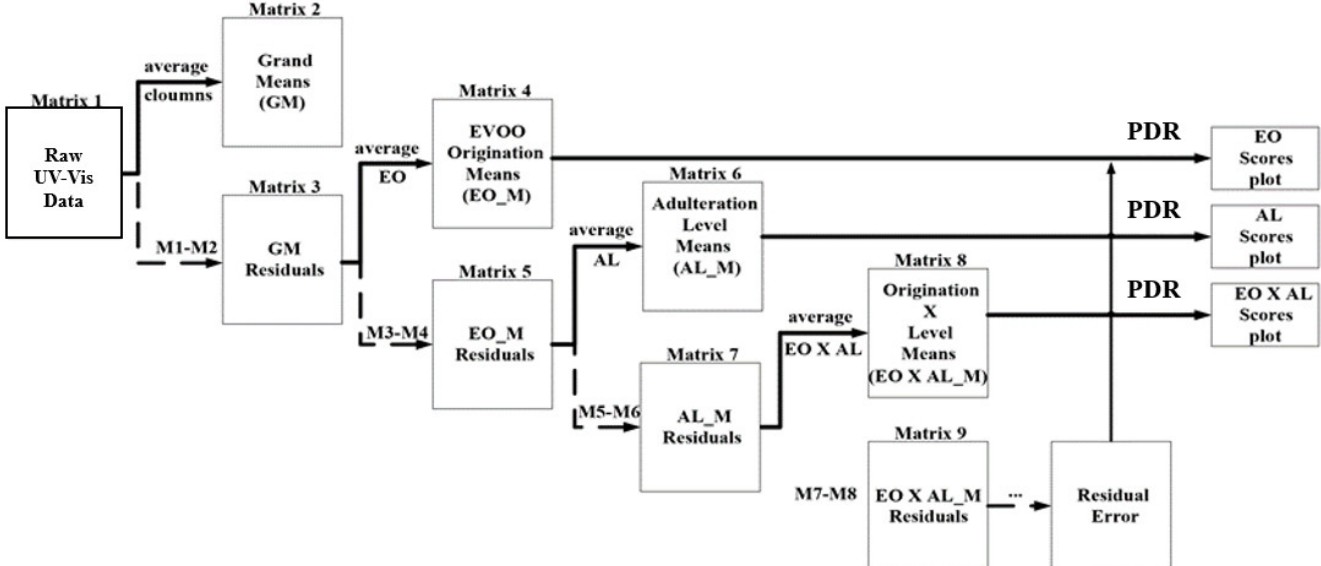

**Figure 1.** Schematic diagram for analysis of variance-projected difference resolution (ANOVA-PDR). The matrix decomposition process was repeated for the other experimental factors and the factor interactions.

By combing the ANOVA and PDR, a straightforward metric for class separation under the influence of multiple factors can be achieved. For each individual factor, the same mean submatrix constructed for ANOVA-PCA were also resolved through PDR according to their respective internal classes. For the main influencing factors, ANOVA-PDR analyzes each class in the corresponding effect matrix in pairs, and expresses it as a triangular matrix to measure the class separation of the data object [19]. To better visualize the separations of various factors between classes, these PDR matrices are plotted in grayscale in this work. The PDRs from small to large corresponds to the color in the color bar from dark to light, so the lighter background color indicates larger PDR that represents a better separation between different classes, and vice versa. The average PDR of each effect matrix was also given to evaluate the factor matrix resolution. The ANOVA-PDR calculation was performed with an in-house script written in MATLAB R2021b (The MathWorks, Natick, MA, USA).

### 2.4. Validation by PLS-DA and PLSR

The PLS-DA and PLSR combined with the bootstrapped Latin partition (BLP) [22] were used to validate quantitative classification and regression models. The dataset is divided into 80% and 20% portions for training and validation using BLP. Nine replicates of adulterated samples were averaged and then used to construct the PLS-DA model, while the pure samples remained unchanged. In PLSR, all 9 repetitions of a same sample were averaged and then used for modeling. In the establishment of classification and regression models, the choice of the number of latent variables is particularly important. In this study, the BLP procedure with 10 bootstraps and 5-fold Latin partitions was used to select the

optimal number of latent variables. The PLS-DA and PLSR validations were calculated by MATLAB in-house scripts (The MathWorks).

## 3. Results

### 3.1. Characteristics of UV-Vis Spectra

Figure 2A–C shows the UV-Vis spectra of the olive oil dataset. Specifically, Figure 2A represents EVOO spectra according to origin; Figure 2B represents spectra according to different levels of adulteration; Figure 2C represents spectra according to different adulteration types. All spectra at the same level to each factor were averaged for presentation purposes. The olive oil samples have multiple absorption peaks in the visible light region. The absorption observed in this spectral region may be dominated by the oil pigments [23]. Specifically, there were three obvious absorption peaks in the 420–480 nm region that correspond to the absorption of blue light by olive oil, which may be mainly related to the carotenoids and chlorophyll contained in olive oil [24]. The peak appearing around 670 nm was also consistent with the absorption of chlorophyll [24]. Therefore, it is interesting to discover whether the pigment compositions can affect the UV-Vis fingerprints by influences of various factors.

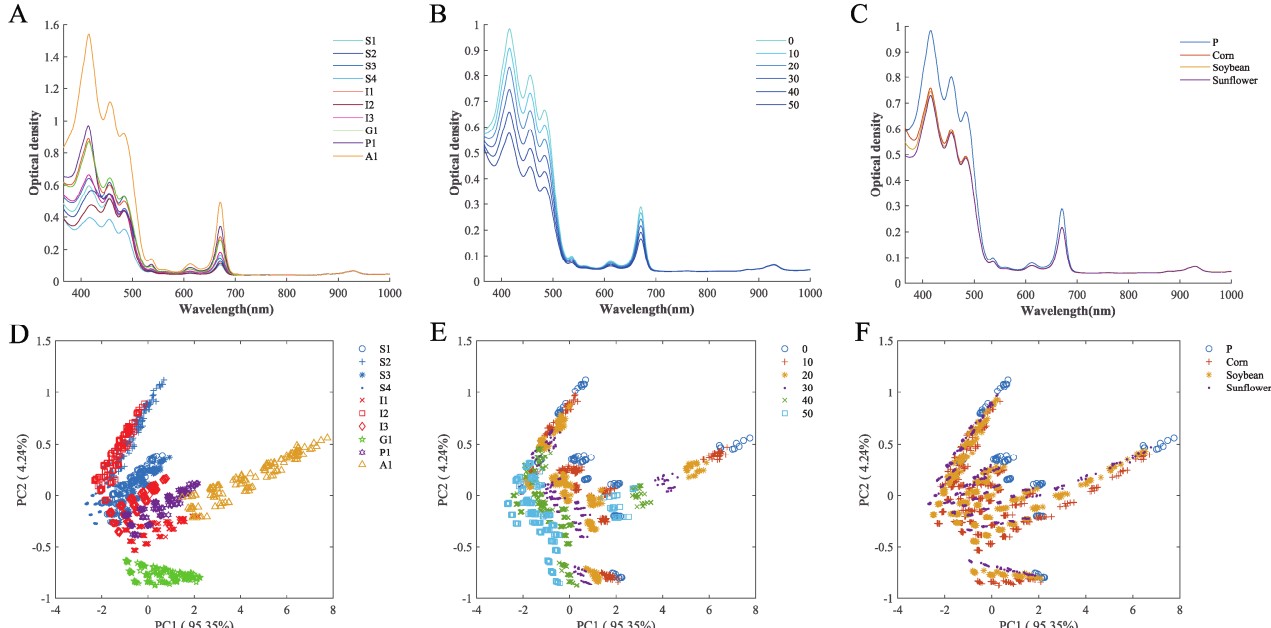

**Figure 2.** Mean UV-Vis spectral profile (**A**–**C**) and conventional PCA score plots (**D**–**F**) of adulterated olive oil according to different factors. (**A**,**D**) EVOO Origin; (**B**,**E**) Adulteration Level; (**C**,**F**) Adulteration Type. EVOO Origin included Spain, S1–S4; Italy, I1–I3; Greece, G1; Portugal, P1; and Australia, A1. In (**A**–**C**), all spectra at the same level were averaged for presentation purposes.

A preliminary study was carried out first by directly observing the original spectra to explore the influence of the three factors of EVOO origin, adulteration level, and adulterated type on the spectra. The spectra were plotted according to these three factors, with different colors designated in each class. Figure 2A shows the UV-Vis average spectrum drawn according to different EVOO origins. There are great differences between different EVOO originations and the biggest difference between A1 (Australia) and other origins. Figure 2B shows an average spectrum according to different adulteration levels. The color in the figure from lighter to darker indicates the increasing degree of adulteration. As the degree of adulteration increases, the absorption peaks of the average spectrum obtained gradually decrease accordingly. Figure 2C shows an average spectrum drawn according to different adulteration types. The difference between pure EVOO and adulterated EVOO is significant, while the differences between different types of adulteration are small. Since

the preliminary observation implied the influences of all factors, it is difficult to obtain individual influences directly. The subsequent chemometric methods will be used to further analyze the three factors.

### 3.2. Direct PCA and PDR

Conventional PCA and PDR were applied to evaluate overall class separations without considering any confounding factors. Figure 2D–F shows the first two largest scores of the UV-Vis spectra using conventional PCA. The score plots are marked by EVOO origin, adulteration level, and adulteration type, so that the impact of these three factors on adulteration spectrum can be easily observed. Figure 3 shows the PDR mapping obtained through conventional PDR processing. The overall degree of discrimination can also be estimated by the average PDRs. Generally, in PDR mapping, each block corresponds to PDRs from any pair of classes, while these classes can be defined as any factors or outputs. The degrees of separations of the entire dataset can be observed in an intuitive way. As a result, the internal separation of the three factors of EVOO origin, adulteration level, and adulteration type are respectively displayed.

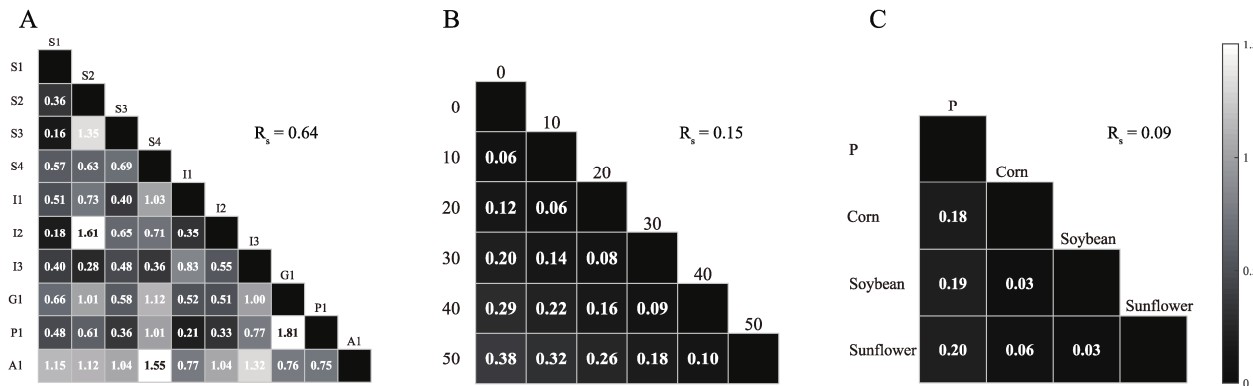

**Figure 3.** Projected difference resolution (PDR) mapping of conventional PDR results. (**A**) EVOO Origin; (**B**) Adulteration Level; (**C**) Adulterant Type. $R_s$, geometric average of all PDR scores. P, pure samples.

As shown in Figure 2E, samples from different EVOO origins were observed to be significantly separated, indicating that the EVOO origin dominated the total variance in the model. Due to the influence of the EVOO origin, there is no significant aggregation between any samples of the same adulteration level or adulteration type, indicating relatively little influences on the UV-Vis spectra. It can be observed from Figure 3 that most PDRs of the three factors are less than 1.5, indicating a poor separation. To sum up, the differences between the producing origin of EVOO significantly affect the UV-Vis spectra. The differences between the adulteration level and the adulteration type remained undetectable, possibly due to the dominating affect by the origin. The conventional PCA and PDR cannot distinguish each factor directly. Therefore, further data treatments by ANOVA to remove the cofounding effects arising from multiple factors are necessary.

### 3.3. ANOVA-PDR

ANOVA-PDR is able to isolate the interferences between factors and to analyze the differences between various classes of interest by each factor. ANOVA-PDR delivered better separation of EVOO origin, as well as adulteration level and adulteration type. The important variables, i.e., spectral peaks, can also be identified by the corresponding loadings. Meanwhile, ANOVA-PDR directly quantifies the distinction between factors within classes. Figure 4 shows the PDR mapping with detailed between-class separations. Compared with the direct PDR in the initial UV-Vis spectrum, ANOVA-PDR resulted in a considerably greater value, thus a clearer distinction between each factor class.

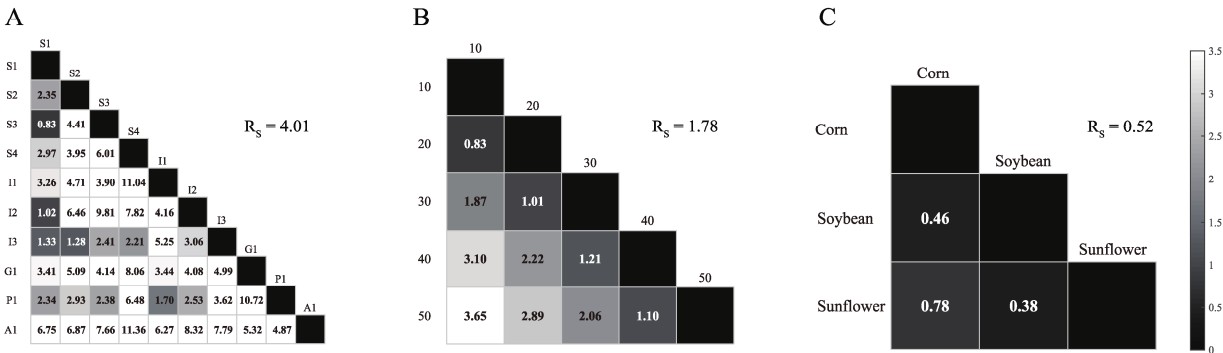

**Figure 4.** ANOVA-PDR mapping of the three main factors. (**A**) EVOO Origin; (**B**) Adulteration Level; (**C**) Adulteration Type. $R_s$, geometric average of all PDR scores.

Detailed relationships of classes and their corresponding influential components can be further analyzed from ANOVA-PDR. Figure 4A corresponds to the ANOVA-derived PDR mapping of the EVOO origin. In this plot, there are significant differences in the samples from different EVOO origins. Specifically, samples A1 produced in Australia were particularly further away from the counterparts produced in other locations, probably due the fact that they were the only olive oils produced in a non-Mediterranean country. Such difference may suggest the concentration of different pigments may be different with respect to originations. Due to the fact that the UV-Vis shows general disadvantages in characterization of compounds, the compositional variations of pigments is rather indicative than exhaustive and complete. Further investigation is necessary. The PDRs between samples increased as the level of adulteration rises, as shown in Figure 4B. Although part of the PDRs between two adjacent adulteration levels was less than 1.5, the average PDR between the concentrations is 1.78, a clear difference. The PDRs suggested that it is foreseeable to establish an effective adulteration detection model, provided that a proper treatment to exclude or reduce the influence from other factors such as EVOO origin is performed. Furthermore, the PDRs values between the various adulteration types in Figure 4C are all less than 1.5, as well as the average PDR of the adulteration types is also less than 1.5, indicating that different adulteration types are difficult to classify. Since all the oils that we used as adulterants were only from China, it may be possible that there are limited variances between oils. Therefore, different types of adulterants carry unnoticeable variances in the overall compositions compared to other factors.

In summary, PDR mappings and average PDR can intuitively and quantitatively conclude that the differences between the classes from large to small are the origin of EVOO, the level of adulteration, and the type of adulteration, reflecting the degree of influence of factors on the adulterated samples. Compared to PCA, PDR is more effective in distinguishing between classes within these three factors.

### 3.4. PLS-DA and PLSR Model Validation

The classification and regression models (PLS-DA and PLSR) were established to further validate class separation under multiple factors. Since the origin of EVOO is the most important source of difference in the UV-Vis spectra of adulterated samples, it may significantly affect the accuracy of the adulteration detection model. Therefore, eight models were established to compare the differences in the sample under various situations. Specifically, the eight models consisted of four groups. The global models included all 10 EVOO producers. The European model excluded the Australian sample, while the Spanish and Italian models only included samples originated from Spain and Italy, respectively. For the three local models, the applicability of the external test set was applied to evaluate the performance of these local models, for which the external test set is defined as the independent test sets that were from the selected countries of origins only.

From the prediction results of the local models in Table 1, it can be observed that the local models generally yielded good performance by using internal training and the test set. The prediction accuracies of the local models by PLS-DA on training and test sets were above 97%. The RMSEs of the PLSR local models on training and test sets were also less than 2.03%. However, these models have a poor predictive effect on the external test sets from other origins. The prediction accuracies of the three PLS-DA local models for these different external test sets dropped to 59.58–87.92%. Meanwhile, the RMSEs of PLSR models also raised significantly. Considering the results of ANOVA-PDR, among the EVOO origin, the difference between A1 and others is the largest. Its RMSE reached 31.39% in the PLSR model of the European model, and lack of quantitative prediction power. Moreover, compared to the external test set of other origins, when A1 is used as the external test set, the RSME of the PLSR model of the Spanish model and the Italian model both increased by more than 50%, and the worst prediction effect is obtained. A more intuitive presentation of the local PLSR model on the external test set from other origins can be observed in Figure 5. In the process of establishing the EVOO adulteration detection model, the EVOO origin is an important influencing factor, but it was typically neglected. For instance, Jiang et al. established PLSR models of EVOO adulterated with corn oil, soybean oil, and sunflower oil by UV-Vis, in which the RSMEs yielded as low as 0.001% [5]. However, these models are based on only one EVOO, so that validation at a larger scale might still be needed.

**Table 1.** EVOO adulteration prediction by UV-Vis spectra according to geographical origins.

| Type | Model | N [a] | LV [b] | Training [c] | Test [c] | Test on Selected Origins [c] | | | | |
|---|---|---|---|---|---|---|---|---|---|---|
| | | | | | | Australia | Greece | Portugal | Italy | Spain |
| PLS-DA | Global | 240 | 18 | 99.95 | 98.12 | N/A | N/A | N/A | N/A | N/A |
| | European | 216 | 15 | 99.77 | 98.60 | 87.92 | N/A | N/A | N/A | N/A |
| | Spanish | 96 | 10 | 99.61 | 97.37 | 62.50 | 62.50 | 67.50 | 85.97 | N/A |
| | Italian | 72 | 7 | 99.48 | 97.14 | 59.58 | 62.50 | 64.17 | N/A | 82.47 |
| PLSR | Global | 160 | 24 | 0.62 | 1.77 | N/A | N/A | N/A | N/A | N/A |
| | European | 144 | 19 | 0.84 | 2.03 | 31.39 | N/A | N/A | N/A | N/A |
| | Spanish | 64 | 13 | 0.50 | 1.44 | 76.86 | 6.81 | 13.40 | 24.59 | N/A |
| | Italian | 48 | 12 | 0.32 | 0.64 | 67.39 | 17.77 | 10.08 | N/A | 14.28 |

[a]: Number of samples in the dataset. [b]: Latent variables used to build the PLS-DA model. [c]: Results are shown as percent prediction accuracy for PLS-DA (rows 1–4), and root mean squared error (RMSE) for PLSR (rows 5–8). N/A, not available.

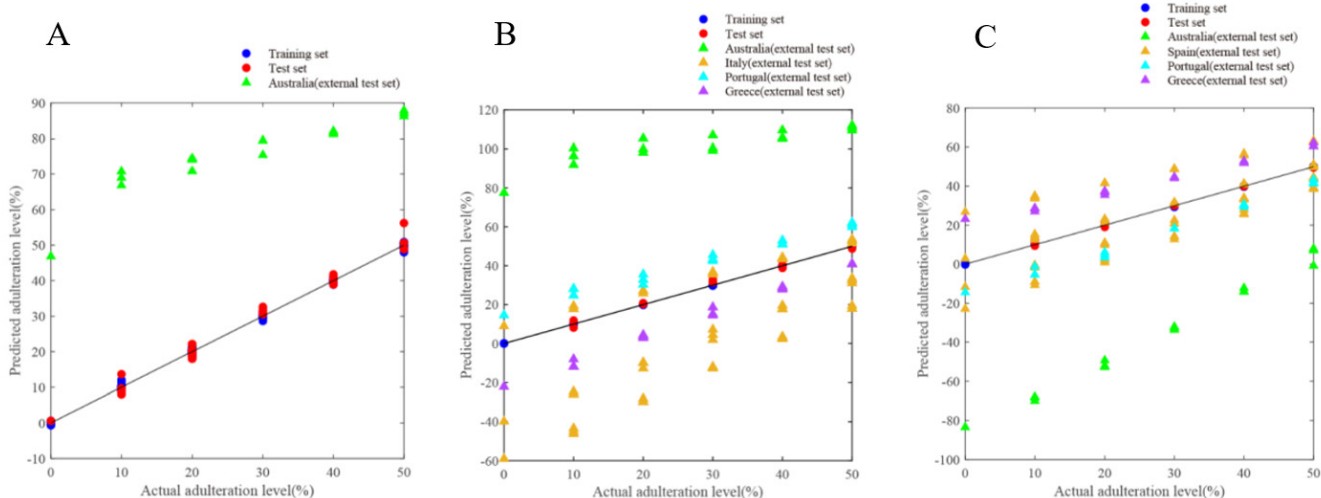

**Figure 5.** Local partial least-squares regression (PLSR) model predictions. (**A**) European. (**B**) Spanish. (**C**) Italian. The black line starts from the origin with a slope of 1, representing the ideal regression result. All predictions from the training, internal, and external test sets are plotted in colored markers in each subfigure for better comparison.

The results of our attempt to establish EVOO adulteration detection models based on different EVOO origins indicated that it may be difficult to establish a universal model for EVOO adulteration detection using UV-Vis spectroscopy. Since multiple EVOO origins, adulteration types, and the interaction between factors are the sources of differences in the original data matrix, all influencing factors must be considered when trying to establish a general model. Otherwise, it is likely that adulteration in the external test set cannot be predicted. The previous studies did not include such variety of samples and the involvement of different influencing factors. On the other hand, although the PLS-DA and PLSR models established by using all data have achieved adequate prediction results with prediction accuracy more than 99% and RSME less than 2%, there may still be issues remaining in the generalization ability of the model. Specifically, with external test sets composed of other EVOO origins or adulterants that are outside the training set, the prediction remains problematic. Additionally, a model with too many latent variables may be prone to overfit, as the model is becoming too complicated. Therefore, it is recommended to build a chemometric model with controlled samples included in the training set for a more accurate prediction.

## 4. Discussion

This study demonstrated that the ANOVA-PDR could be a valuable tool for UV-Vis spectroscopy to identify the sources of variations in a complicated sample set from multi-factorial-designed experiments. The ANOVA was combined with PDR for the first time and provided an exact and comprehensive comparison of the differences between classes and offered results with visual plots, which helps interpret the significance for the arrangement and control of factors. The overall degrees of separation are evaluated by calculating the geometric mean of the PDRs. The ANOVA-PDR was proved to be an effective supplement to multivariate modeling such as PLS-DA.

With respect to the UV-Vis spectroscopy of olive oil adulteration, the study indicated that the EVOO origin and adulteration level are effective sources of variation in the spectra, which may cause potential difficulties in the suitability of the EVOO adulteration detection model. The subsequent PLS-DA and PLSR models for EVOO adulteration detection were also consistent with this conclusion. The results demonstrated that the EVOO adulteration detection model established by the UV-Vis spectroscopy combined with PLS may achieve unbiased results without the aid of a proper model transfer and validation routine. To overcome this problem, further research can also focus on a more sensitive and selective detection methods, as well as to devise a controlled approach to select training samples for the chemometrics model. Additionally, since all the adulterants in this study were collected from China only, the adulterant-type factor raised insignificant variances. When analyzing possible impurities, it is better to examine both adulterant oils (corn, soy, sunflower) from local producers, as well as producers from the countries of origin of olive oil. In addition, other types of widely available oils, such as canola and peanut oils could be included in the model. In this manner, a robust and reliable model can be achieved.

**Author Contributions:** Conceptualization and funding acquisition, B.G.; methodology and investigation, J.Z.; validation, W.L.; writing—original draft preparation, J.Z.; writing—review and editing, W.L. All authors have read and agreed to the published version of the manuscript.

**Funding:** This research was funded by The National Natural Science Foundation of China (Grant No. 32001819, 32272426).

**Institutional Review Board Statement:** Not applicable.

**Informed Consent Statement:** Not applicable.

**Data Availability Statement:** The data presented in this study are openly available upon request.

**Conflicts of Interest:** The authors declare no conflict of interest.

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
