# Peer review of "Characterizing Variances of Adulterated Extra Virgin Olive Oils by UV-Vis Spectroscopy Combined with Analysis of Variance-Projected Difference Resolution (ANOVA-PDR) and Multivariate Classification"

_applsci, doi:10.3390/app13074360_

Round 1
Reviewer 1 Report
My comments are included into pdf file.
Please be so kind and state the novelty of your scientific research.

Author Response
Itemized responses to reviewers’ comments
Manuscript ID: applsci-2270430
Title: Characterizing variances of adulterated extra virgin olive oils by UV-Vis spectroscopy combined with projected difference resolution (ANOVA-PDR) and multivariate classification
Reviewer 1
Comments:
- Please be so kind and state the novelty of your scientific research.
Response: The novelty of this study is that this study proposed a novel method, analysis of variance-projected difference resolution (ANOVA-PDR), for detecting EVOO adulteration while considering multiple influencing factors, including origin, adulteration level, and adulteration type. (Line 95-97)
- Abstract: Clearly emphsize the aim of the study and put some important results obtained in your work. This abstract is too general.
Response: Done. Thank you for your comment. We have revised the abstract accordingly (Line 10-20). It is now given as the following:
The analysis of variance-projected difference resolution (ANOVA-PDR) were proposed and compared with multivariate classification for its potential in detecting possible food adulteration in extra virgin olive oils (EVOOs) by UV-Vis spectra. Three factors including origin, adulteration level, and adulteration type were systematically examined by the ANOVA-derived methods. The ANOVA-PDR quantitatively presented the separation of the internal classes according to the three main factors. Specifically, the average ANOVA-derived PDRs of the EVOO origination and adulteration level respectively is 4.01 and 1.78, while the conventional PDRs of the three factors are all less than 1.5. Furthermore, the partial least-squares-discriminant analysis (PLS-DA) and the PLS regression (PLSR) modelling with the selected sub-datasets from different origins to verify the results. The resulting models suggested that the three main factors and their interactions were all important sources of spectral variations.
- Introduction, Line 35-36:”According to previous studies, the UV-Vis spectroscopy is one of the important techniques for the adulteration detection, authentication of the geographic location or the grade of a specific olive oil product.”
Comment: This part of Introduction should be extended in a way to link UV-Vis spectroscopy together with chemometrics approacH. It is not enough to say that based on UV-Vis spectra you can predict something; based on UV-Vis spectra in combination with PCA, PLS, ANOVA-PDR....
Response: Done. We have revised this part accordingly, stating that the UV-Vis spectroscopy combined with chemometrics is one of the important techniques for the adulteration detection, and so on. (Line 36-37)
- Introduction, Line 37-38 “quantified the level of adulteration in Spanish EVOO from their UV-Vis spectra [3, 4].”
Comment: How did the quantify the level of adulteration ?
Response: Done revising. The level of adulteration was quantified using linear and non-linear modeling based on 17 chaotic parameters calculated by UV-vis scans. Linear models with more independent variables showed better statistical results. A radial basis network model with one input node and one output neuron was used for non-linear modeling. The manuscript has also been revised and this information added accordingly. Corresponding statements were added at Line 40-43.
- Introduction, Line 42-43 “The UV-Vis spectroscopy has been used to determine the geographic origin of EVOO as well [8, 9].”
Comment: How did they connect UV-Vis spectroscopy with the geographic origin of EVOO ??
Response: Done. We have added relevant and detailed descriptions to this revision (Line 48-56). It is given as the following:
In these studies, UV-Vis spectroscopy and high-performance liquid chromatography with diode array detector were used to quantify main pigments in several EVOOs and compared the advantages and disadvantages of both techniques. The methods were applied to a selection of monovarietal EVOOs produced in different geographical areas in Mediterranean countries. The differences among EVOOs produced in different geographic areas were analyzed using principal component analysis (PCA) and independent component analysis to evaluate the correlation between pigments' content such as chlorophylls and carotenoids in olive oils and factors such as ripeness stage, geographic origin, and cultivars.
- Introduction, Line 53 “Multi-influencing factors can have a significant ……
Comment: This paragraph should be also rewritten in a more concise way. The emphasize should be on adulteration of EVOO, factors influencing quality of the EVOO and development of models capable for predicting adulteration based on UV-Vis spectra.
Response: Done. We have rewritten the entire paragraph in a more concise way (Line 67-73). It is now given as the following:
Multi-factors can significantly impact the chemical analysis procedure, such as ac-curacy, sensitivity, and reproducibility. In the case of EVOO adulteration detection, factors like origin, adulteration level, and type of the adulterant can affect the robustness of the model. Analyzing the relationships between these factors can guide the establishment of subsequent detection models and evaluate their significance. Techniques to analyze the influence of multiple factors are highly desirable for accurate analysis of EVOOs from different manufacturers.
- Introduction, Line 62 “… EVOO adulteration detection model …”
Comment: Are you planning to develop EVOO adulteration detection model ?? This sentence should be rewritten.
Response: We appreciate your valuable input, and apologize for any confusion caused by the initial version. To clarify, we are not intended in developing an EVOO adulteration detection model specifically in this work. However, we believe that analyzing the relationships between multiple factors can guide the establishment of subsequent detection models in food analysis (Line 70-71).
- Introduction, Line 94 “In this work, ANOVA-PDR were proposed and applied …”
Comment: This last parapgraph should be written in a more concise way. Clearly written what was the aim of the study: to develop method for detection of EVOO adulteration taking into account factors (mention them). Then mention validation of the developed model (PLS-DA, PLSR).
Response: Done. We have rewritten the entire paragraph in a more concise way, focusing the aim of this study (Line 95-102). It is now given as the following:
The aim of this study was to propose a novel method, analysis of variance-projected difference resolution (ANOVA-PDR), for detecting EVOO adulteration while considering multiple influencing factors, including origin, adulteration level, and adulteration type. The UV-Vis spectra of adulterated EVOOs were comprehensively analyzed using ANO-VA-PDR techniques, and the results were validated using PLS-DA and PLSR to build both quantitative and qualitative adulteration models. ANOVA-PDR can evaluate modelling performance in relation to the multiple sampling factors of the EVOO adulteration detection model.
- Materials and Methods, section 2.3 ANOVA-PDR, header at Line 132.
Comment: software, version, manufacturer
Response: Done revising. Added “All data processing algorithms, including data pre-processing, ANOVA-PDR, PLS-DA, and PLSR, were developed and performed on a personal computer with in-house scripts written in MATLAB R2021b (The MathWorks, Natick, Massachusetts, USA).” (Line 131-133)
- Materials and Methods, Line 139 “… for each experimental factor …”
Comment: mention experimental factor(factors. Unclear ! What was the output variables ??
Response: Done. Specific experimental factors were described in detail as: “Specifically, the factors of this EVOO adulteration study included origin, adulteration level, and adulteration type.” (Line 140-141). The output variables, in terms of this EVOO adulteration study, would be the final UV-Vis spectra. Factor may be confusing, therefore, we defined it in Line 56-57 (For brevity, “experimental factor” is addressed as factors for all subsequent descriptions.) for better clarity.
- Figure 1, Captions, “ANOVA-PDR” Line 144
Comment: full title of the ANOVA-PDR; Fig 1 is of low quality; raw UV-VIs data (add to box Matrix 1)
Response: Done. Full title of the ANOVA-PDR given in the figure caption. (Line 147-149) Due to some unknown problem, the pdf version always shows a low-quality figure, no matter the resolution in our original Word document. Therefore, high-resolution figures were also independently given in the end of the manuscript. Matrix 1 has also been revised accordingly.
- Results, Line 187 “Figure 2 (A-C) is the UV-Vis spectra of adulterated olive oil.”
Comment: Is this true ?? Check the second parapgraph. I believe that Figure 2A represents EVOO spectra according to origin, Figure 2B represents different levels of adulteration a Figure 2C represents adulteration types.
Response: You are correct. We appreciate you for pointing out this mistake. We have revised and corrected in this revision, with extensive explanations added to avoid misunderstanding, as: “Figure 2 (A-C) is the UV-Vis spectra of the olive oil data set. Specifically, Figure 2A represents EVOO spectra according to origin; Figure 2B represents spectra according to different levels of adulteration; Figure 2C represents spectra according to different adulteration types. All spectra at the same level to each factor were averaged for presentation purposes.” (Line 192-196)
- Figure 2 caption, Line 195
Comment: low quality of the figures and legends
Response: Done. High-resolution figures were given in the end of the manuscript.
- Figure 3 caption, Line 236
Comment: low quality of figure. Explain abbreviations in the figure caption. Add what is Rs.
Response: Done. High-resolution figures were given in the end of the manuscript. Abbreviations explained in the figure caption, including Rs explained being average PDR score (Line 248-250). Thank you for this valuable comment.
- Figure 4 caption, Line 248
Comment: low figure quality
Response: Done. High-resolution figures were given in the end of the manuscript.
- Results, Line 264, “… a universal quantitative adulteration detection model …”
Comment: Are you sure ? Provide us with some additional information that support this fact. Is there some literature that is in agreement with your results?
Response: We think that the original statement “establish a universal quantitative adulteration detection model” is beyond the scope of this research and inappropriate. It is revised to “to establish an effective adulteration detection model” for clarity (Line 276-277). Thank you for your comment.
- Results, Line 275, “… with the results of pooed-ANOVA and reflect the degree …”
Comment: ?
Response: Done. This is an error. We removed it in this revision (Line 287).
- Results, Line 287 “… the external test set …”
Comment: validation
Response: Thank you for addressing it out. In chemometrics terms, a validation set and a test set are both subsets of a dataset used to evaluate the performance of a predictive model. They may be used interchangeably in many manuscripts. However, the validation set is typically used for model selection and hyperparameter tuning. It is a subset of the original dataset that is used to tune the model's parameters and evaluate its performance. The test set, on the other hand, is a subset of the dataset that is used to evaluate the performance of the final model. It is a completely independent dataset that the model has not used during training or validation. The purpose of the test set is to provide an unbiased estimate of the model's performance on new, unused data.
Generally, “validation” is for internal test set, such as for cross-validation, “test” is for external test set. As the test set used here is for evaluate performance of the final models, therefore, we keep this term unchanged to follow this convention in this revision.
- Table 1, Line 289 “Table 1. Prediction of EVOO adulteration by PLS-DA and PLSR models with selected data sets defined according to origins.”
Comment: improve the title of the table
Response: Done. The title of the table is revised as: “EVOO adulteration prediction by UV-Vis spectra according to geographical origins” (Line 301).
- Table 1, Line 290
Comment: You excluded Australia from european model?
Response: Yes. It is specifically designed and described in the first paragraph of Section 3.4 (Line 297-300).
- Table 1, Line 293 “c: Results were shown as percent prediction accuracy and root mean squared error (RMSE) for PLS-DA and PLSR, respectively.”
Comment: unclear. Specify in the table what is percente prediction accuray and what is RMSE.
Response: Done. Considering the wide spread usage of percent prediction accuracy and RMSE in chemometrics, we revised and added explanation in a brief way as: “Results were shown as percent prediction accuracy, i.e., the percentage of correctly predicted samples in the specific test set, and root mean squared error (RMSE), i.e., the square root of the average of the squared differences between the predicted and actual values, for PLS-DA and PLSR, respectively.” (Line 304-307).
- Figure 5 caption, Line 296.
Comment: low quality figure. Cannot see black line. Figure caption should be better explain
Response: Done. High-resolution figures were given in the end of the document. Figure captions were revised as: “Local PLSR model predictions. (A) European. (B) Spanish. (C) Italian. The black line starts from the origin with a slope of 1, representing the ideal regression result. All predictions from the training, internal, and external test sets were plotted in colored markers in each subfigure for better comparison.” (Line 309-312)
- Results, Line 306 “87.92%”.
Comment: where is this number in table?
Response: This number, 87.92%, is the prediction accuracy of the European model when using Australia EVOOs as external test set. (Table 1 at Line 301, second row, first column in external test set) It resembles the highest prediction accuracy possible when using local models for different external test sets. However, it is still worse than any internal test set.
Reviewer 2 Report
This manuscript documents an attempt to use ANOVA-PDR to determine if adulteration of extra virgin olive oil by lesser value oils could be unambiguously detected by UV-VIS spectroscopy. The work shows that a general model is not effective since the geographical origin of the olives has a significant effect on the variability in the spectra of these oils. Some issues do need to be addressed before publication can be recommended.
All of the figures need to be clearer, as many of them were too blurry to read and understand. Higher resolution of the graphs and or better color selections to clearly identify individual samples are definitely required.
As the olive oils from the different countries were purchased from a grocery store, it is not clear that what was purchased was not already adulterated. How do the authors ensure that the starting materials are pure and exactly what they say that they are?
In equation 2, what is the term X(subI)(superT)
Line 189 “maybe” should be “may be”
The spectra of the contaminates should also be reported so that readers can understand what might be observed in the spectra that are due to the corn, sunflower or soybean oils. Because canola oil is the most widely available vegetable oil, it is not clear why this was not also evaluated or at least commented on.
In Figure 2, panel B shows a change in spectra due to changes in the amount of adulteration, but the legend does not list which of the 10 olive oils was being evaluated in this plot, or was this spectra actually an average spectra of all 10 of the olive oils? Also, which of the adulterating oils was used, or again, was this an average of all data for all three adulterants? Please be clear in what is being reported in the figures.
This same issue is also true of Figure 2C where spectra taken with the three different adulterant oils is shown. Again, was this for a single olive oil at a single adulterant concentration? Make sure the reader understands what is being reported.
Line 217 should read “Figure 2 (D-F)
Line 220 “Figure 2” should be “Figure 3”
In the discussion of Figure 4A, Lines 252-254, the authors state: “In this plot there are significant differences in the adulterated samples from different EVOO origin.” This figure, however, is showing results from unadulterated oils, just looking at the differences that are observed strictly due to country/region of origin.
Figures 3B and 3C, as well as 4B and 4C also have to be better detailed. Again, it is not clear if these are combinations of data from all of the oils/adulteration conditions, or simply from a single oil and set of conditions.
In line 269, the authors state: “The result is consistent with the fact that the presence of adulterants was in low concentrations in this study” This is not a correct statement. Having an adulterant concentration of 50% is not a low concentration.
Line 276 states: “Compared to PDR, PCA is more effective in distinguishing between classes within these three factors.” It is unclear why the authors can make this claim, as they have shown no real results that PCA can better distinguish adulterants than the PDR results.
Figure 5C is also concerning. This model, developed using only the Italian oil results, shows negative adulteration for the Australian oil while both the European and Spanish models show positive adulteration values for this oil. While there are fewer latent variables in the model using only oils from Italy, the reversal of the Australian oil analysis suggests that there are issues with the model that was developed.
Line 319 should read “might still be needed.
Author Response
Itemized responses to reviewers’ comments
Manuscript ID: applsci-2270430
Title: Characterizing variances of adulterated extra virgin olive oils by UV-Vis spectroscopy combined with projected difference resolution (ANOVA-PDR) and multivariate classification
Reviewer 2
This manuscript documents an attempt to use ANOVA-PDR to determine if adulteration of extra virgin olive oil by lesser value oils could be unambiguously detected by UV-VIS spectroscopy. The work shows that a general model is not effective since the geographical origin of the olives has a significant effect on the variability in the spectra of these oils. Some issues do need to be addressed before publication can be recommended.
All of the figures need to be clearer, as many of them were too blurry to read and understand. Higher resolution of the graphs and or better color selections to clearly identify individual samples are definitely required.
Response: Done. Due to some unknown problem, the pdf version always shows a low-quality figure, no matter the resolution in our original Word document. Therefore, all high-resolution figures were directly uploaded and given in the end of the manuscript.
As the olive oils from the different countries were purchased from a grocery store, it is not clear that what was purchased was not already adulterated. How do the authors ensure that the starting materials are pure and exactly what they say that they are?
Response: Thank you for your comment. Due to the limitation of experimental resources, we choose commercial sources of samples. However, we choose to collect as many varieties of sources (many stores, brands, suppliers, instead of one grocery store). Also, the authenticity of EVOOs were also warranted by local importers according to national regulations of imported olive oils, with names and detailed contacting information listed. On the other hand, the primary goal of our study was to demonstrate propose a novel method, analysis of variance-projected difference resolution (ANOVA-PDR), for detecting EVOO adulteration while considering multiple influencing factors. Even if there were some falsified samples, the chemometrics model will rule out these unrelated interferences and model the factor we want.
- In equation 2, what is the term X(subI)(superT)
Response: X(subI)(superT) is the transposed two-way matrix for each target class. i could be class a or b in this case. Relevant statements were added at Line 158-159.
- Line 189 “maybe” should be “may be”
Response: Revised as suggested (Line 197). Thank you.
- The spectra of the contaminates should also be reported so that readers can understand what might be observed in the spectra that are due to the corn, sunflower or soybean oils. Because canola oil is the most widely available vegetable oil, it is not clear why this was not also evaluated or at least commented on.
Response: We agree that it is an excellent idea to present the reader the spectra of the contaminates. However, we did not record it during the experimental phase of the study. It is suspected for now to record the spectra since the spectra may alter due to possible oxidation as time passing by. If a novel batch of sample collected, it is also suspicious that batch-to-batch variation may lead to another experimental factors. Therefore, we have to decide to make no changes.
On the other hand, the oils selected in this study is limited in a small number because as a pilot study, increasing type of adulterants will cause a large increase to number of overall samples being evaluated. However, we commented on selection of oils as: “Also, other types of widely available oils, such as canola and peanut oils could be included in the model. In this manner a robust and reliable model can be achieved.” (Line 372-374)
- In Figure 2, panel B shows a change in spectra due to changes in the amount of adulteration, but the legend does not list which of the 10 olive oils was being evaluated in this plot, or was this spectra actually an average spectra of all 10 of the olive oils? Also, which of the adulterating oils was used, or again, was this an average of all data for all three adulterants? Please be clear in what is being reported in the figures.
Response: Revised as suggested. To clarify, all spectra at the same level to each factor were averaged for presentation purposes (Line 195-196, Figure 2 captions at Line 208).
- This same issue is also true of Figure 2C where spectra taken with the three different adulterant oils is shown. Again, was this for a single olive oil at a single adulterant concentration? Make sure the reader understands what is being reported.
Response: Revised as suggested as the same for previous questions (Line 195-196, Figure 2 captions at Line 208). We thank the reviewer for this comment.
- Line 217 should read “Figure 2 (D-F)
Response: Thank you for pointing this out. Revised as suggested. (Line 226)
- Line 220 “Figure 2” should be “Figure 3”
Response: Thank you. Revised. (Line 229)
- In the discussion of Figure 4A, Lines 252-254, the authors state: “In this plot there are significant differences in the adulterated samples from different EVOO origin.” This figure, however, is showing results from unadulterated oils, just looking at the differences that are observed strictly due to country/region of origin.
Response: Thank you for your comment. To clarify, samples, whether adulterated or unadulterated, were evaluated for their ANOVA-PDR according to the origin. We are not sure where the authors can obtain specific information that these samples were all unadulterated, and welcome the reviewer to comment further to improve the expression. For now, we removed “adulterated” in the said sentence for better clarity (Line 266-267).
- Figures 3B and 3C, as well as 4B and 4C also have to be better detailed. Again, it is not clear if these are combinations of data from all of the oils/adulteration conditions, or simply from a single oil and set of conditions.
Response: Thank you for your comment. Conventional PDR mapping is treating one factor at a time, that corresponding to the variances of the entire data set, which is combinations of data from all of the oils/adulteration conditions. To help better understand PDR mapping, the following description is given at Line 231-233: Generally, in PDR mapping, each block corresponds to PDRs from any pair of classes, while these classes can be defined as any factors or outputs. The degrees of separations of the entire data set can be observed in an intuitive way.
- In line 269, the authors state: “The result is consistent with the fact that the presence of adulterants was in low concentrations in this study” This is not a correct statement. Having an adulterant concentration of 50% is not a low concentration.
Response: We appreciate and agree with your comment. This sentence is removed, as a result. (Line 280-281)
Line 276 states: “Compared to PDR, PCA is more effective in distinguishing between classes within these three factors.” It is unclear why the authors can make this claim, as they have shown no real results that PCA can better distinguish adulterants than the PDR results.
Response: Thank you for your comment again. This statement has an error in it. We corrected it as: “Compared to PCA, PDR is more effective in distinguishing between classes within these three factors” (Line 288-289).
Figure 5C is also concerning. This model, developed using only the Italian oil results, shows negative adulteration for the Australian oil while both the European and Spanish models show positive adulteration values for this oil. While there are fewer latent variables in the model using only oils from Italy, the reversal of the Australian oil analysis suggests that there are issues with the model that was developed.
Response: The negative adulteration, not only in Figure 5C, but also in Figure 5B, indicate a poor, or even totally failing predictive effect on the external test sets from other origins as stated at Line 318-319. This issue that arises the reviewer’s concern, on the contrary, is just the evidence that we should apply technologies such as ANOVA-PDR to systematically evaluate the factor influence, as we further analyze this negative effect in the subsequent results (Line 319-349). We want to mention that building an effective adulteration detection model is not our primary goal. Using analysis of variance-projected difference resolution (ANOVA-PDR), for evaluating EVOO adulteration while considering multiple influencing factors, including origin, adulteration level, and adulteration type, is our major aim in this study. The PLS-DA and PLSR just demonstrated some extreme cases without the analysis of experimental factors.
Line 319 should read “might still be needed.
Response: Revised (Line 333). Thank you.
Reviewer 3 Report
In the presented Manuscript, the authors touched upon the rather topical topic of food adulteration (in particular, olive oil) and the development of methods for its determination using UV-visible spectroscopy.
The following comments and recommendations are available on this Manuscript:
1. Lines 112-113. When analyzing possible impurities, it is possible to examine both oil (corn, soy, sunflower) from local producers and producers from the countries of origin of olive oil. It depends on the place of possible falsifications: the country of production or the country of sale. In future studies, I recommend that the authors pay attention to this.
2. Lines 127-128: It is necessary to clarify the method of measuring spectra, describing it in more detail.
3. Figures 1, 2, 5 are of very poor quality. This does not allow us to verify the authors' conclusions.
Based on the review of the Manuscript, I consider this study by the authors on the diagnosis of adulteration of olive oil preliminary and in need of more extensive experimental studies.
Author Response
Itemized responses to reviewers’ comments
Manuscript ID: applsci-2270430
Title: Characterizing variances of adulterated extra virgin olive oils by UV-Vis spectroscopy combined with analysis of variance- projected difference resolution (ANOVA-PDR) and multivariate classification
Reviewer 3
In the presented Manuscript, the authors touched upon the rather topical topic of food adulteration (in particular, olive oil) and the development of methods for its determination using UV-visible spectroscopy.
The following comments and recommendations are available on this Manuscript:
- Lines 112-113. When analyzing possible impurities, it is possible to examine both oil (corn, soy, sunflower) from local producers and producers from the countries of origin of olive oil. It depends on the place of possible falsifications: the country of production or the country of sale. In future studies, I recommend that the authors pay attention to this.
Response: Thank you for your valuable comment. We have added two corresponding statements in the manuscript to make the discussions better:
“Since all the oils that used as adulterants were only from China, it may be possible there is limited variances between oils (Line 281-283).”
“Additionally, since all the adulterants in this study were collected only from China, the adulterants type factor raised insignificant variances. When analyzing possible impurities, it is better to examine both adulterant oils (corn, soy, sunflower) from local producers, as well as producers from the countries of origin of olive oil. … … In this manner a robust and reliable model can be achieved (Line 368-373).”
- Lines 127-128: It is necessary to clarify the method of measuring spectra, describing it in more detail.
Response: The necessary instrument types and settings were given in the original manuscript. As UV-Vis spectroscopy is a relatively simple and straightforward measurement, there is not too much parameters reported. For more additional descriptions and performance comparisons, the reviews are welcome to read reference 10: He, H. and W. Lu, High-throughput chemometric quality assessment of extra virgin olive oils using a microtiter plate reader. Sensors, 2019. 19(19).
We also welcome the reviewer to address any specific information that needed to be added in this section to further clarify the reviewer’s concerns. Therefore, no changes were made right now.
- Figures 1, 2, 5 are of very poor quality. This does not allow us to verify the authors' conclusions.
Response: Done. Due to some unknown problem, the pdf version always shows a low-quality figure, no matter the resolution in our original Word document. Therefore, all high-resolution figures were directly uploaded and given in the end of the manuscript.
- Based on the review of the Manuscript, I consider this study by the authors on the diagnosis of adulteration of olive oil preliminary and in need of more extensive experimental studies.
Response: Thank you. We agree that this is a preliminary study of detecting adulterated extra virgin olive oils. However, building an effective adulteration detection model is not our primary goal. Using analysis of variance-projected difference resolution (ANOVA-PDR), for evaluating EVOO adulteration while considering multiple influencing factors, including origin, adulteration level, and adulteration type, is our major aim in this study. Therefore, EVOO adulteration is selected as a typical, exemplified model, as it shares importance in the food industry, and its typical challenge of adulteration as well as origin identification, with the important traits of multiple influencing factors. The PLS-DA and PLSR just demonstrated some extreme cases without the analysis of experimental factors. Overall, we believe the ANOVA-PDR could be valuable tools for UV-Vis spectroscopy in identifying food adulteration.
Round 2
Reviewer 1 Report
Dear authors,
I have inserted some comments into pdf version. Please be so kind and provide us with responses.

Author Response
Itemized responses to reviewers’ comments
Manuscript ID: applsci-2270430
Title: Characterizing variances of adulterated extra virgin olive oils by UV-Vis spectroscopy combined with analysis of variance- projected difference resolution (ANOVA-PDR) and multivariate classification
Reviewer 1
Comments:
- Materials and Methods, Line 131-133: “All subsequent data processing algorithms was developed and performed on a personal computer with in-house scripts written in MATLAB R2021b (The MathWorks, Natick, Massachusetts, USA)”
Comment: I suggest an additional subchapter for this part since you used one software for ANOVA-PDR, PLS-DA and PLSR. Or you can add in subsections 2.3 and 2.4 (with slight modification of this sentence).
Response: Done accordingly. Thank you for your comment. This statement was added in subsections 2.3 and 2.4 with slight modifications to make it more specific and accurate, as now given in Line 177-179:
“The ANOVA-PDR calculation was performed with an in-house script written in MATLAB R2021b (The MathWorks, Natick, Massachusetts, USA).”
And in Line 189-190:
“The PLS-DA and PLSR validations were calculated by MATLAB in-house scripts (The MathWorks).”
- Results, Line 291: The classification and regression models
Comment: mention PLS-DA and PLSR....The classification and regression models (PLS-DA and PLSR)....
Response: Done. Thank you for your comment. Revised accordingly as: The classification and regression models (PLS-DA and PLSR).... (Line 292)
- Results, Line 291: based on the partial-least squares modelling
Comment: you can remove this part
Response: Done. Removed. (Line 292)
- Table 1 caption
Comment: you should specify which column is RMSE and which column is percent prediction accuracy. This is still unclear. There is no need to put the definition of RMSE.
Response: Done. We appreciate your comment. For a clearer presentation, we revised the footnote of Table 1, removed the definition of RMSE, and specify which part is RMSE and which part is percent prediction accuracy as:
“c: Results were shown as percent prediction accuracy for PLS-DA (rows 1-4), and root mean squared error (RMSE) for PLSR (rows 5-8).” (Line 307-308)
- Table 1 column header
Comment: Although you have explained the term Test and External test this is still a confusing in Table 1. This is connected with my comment aatached to Table 1.
Response: Done. To avoid further confusion, the term “External test” were replaced by “Test on selected origins” in Table 1 column header at Line 304.
Other changes were made to define the term “external test” that make the external test in the main text clearer and easier to understand:
“…, for which the external test set is defined as the independent test sets that were from the selected countries of origins only.” (Line 301-302)
- Figure 5 caption, Line 309: Local PLSR model predictions…
Comment: Please provide full name and then abbreviation
Response: Done. Revised accordingly as: “Local partial least-squares regression (PLSR) model predictions….” (Line 310) We appreciate your comment.